# Senotherapeutics in Cancer and HIV

**DOI:** 10.3390/cells11071222

**Published:** 2022-04-04

**Authors:** Laura Sánchez-Díaz, Asunción Espinosa-Sánchez, José-Ramón Blanco, Amancio Carnero

**Affiliations:** 1Instituto de Biomedicina de Sevilla, IBIS, Hospital Universitario Virgen del Rocío, Universidad de Sevilla, Consejo Superior de Investigaciones Científicas, 41013 Seville, Spain; lsanchez-ibis@us.es (L.S.-D.); aespinosa-ibis@us.es (A.E.-S.); 2CIBERONC, Instituto de Salud Carlos III, 28029 Madrid, Spain; 3Hospital San Pedro/Centro de Investigación Biomédica de La Rioja (CIBIR), 26006 Logroño, Spain; jrblanco@riojasalud.es

**Keywords:** senescence, senotherapy, cancer, HIV, senolytic, senostatic, SASP

## Abstract

Cellular senescence is a stress-response mechanism that contributes to homeostasis maintenance, playing a beneficial role during embryogenesis and in normal adult organisms. In contrast, chronic senescence activation may be responsible for other events such as age-related disorders, HIV and cancer development. Cellular senescence activation can be triggered by different insults. Regardless of the inducer, there are several phenotypes generally shared among senescent cells: cell division arrest, an aberrant shape, increased size, high granularity because of increased numbers of lysosomes and vacuoles, apoptosis resistance, defective metabolism and some chromatin alterations. Senescent cells constitute an important area for research due to their contributions to the pathogenesis of different diseases such as frailty, sarcopenia and aging-related diseases, including cancer and HIV infection, which show an accelerated aging. Hence, a new pharmacological category of treatments called senotherapeutics is under development. This group includes senolytic drugs that selectively attack senescent cells and senostatic drugs that suppress SASP factor delivery, inhibiting senescent cell development. These new drugs can have positive therapeutic effects on aging-related disorders and act in cancer as antitumor drugs, avoiding the undesired effects of senescent cells such as those from SASP. Here, we review senotherapeutics and how they might affect cancer and HIV disease, two very different aging-related diseases, and review some compounds acting as senolytics in clinical trials.

## 1. Biology and Molecular Mechanisms of Senescence

### 1.1. Definition and Induction of Cellular Senescence

Cellular senescence was first described by Hayflick and Moorhead in 1961 [1] in normal human embryonic lung fibroblasts showing irreversible cell division arrest in vitro (replicative senescence) and was subsequently demonstrated in several other cell types. Cellular senescence is a stress-response mechanism that contributes to homeostatic maintenance, playing a beneficial role during embryogenesis and in normal adult organisms. In contrast, chronic activation of senescence may be responsible for other events such as age-related disorders, human immunodeficiency virus (HIV) induced immune system exhaustion and cancer progression/relapse.

Cellular senescence can be triggered by a range of stress stimuli. The first senescence mechanism discovered was gradual telomere shortening in proliferating cells (telomere dependent) [2,3], which impaired cell division. This type of senescence is known as replicative senescence. Telomere erosion serves as a mitotic clock, inducing cell senescence, whose activation prompts normal cells to enter a state of proliferative arrest. This replicative senescence does not occur in stem cells or tumour cells since they express elevated levels of telomerase, endowing them with unlimited replicative potential [4,5]. Other types of cellular senescence include programmed senescence (developmental senescence), which is essential in embryonic development, and stress-induced premature senescence (SIPS) [6], which can be activated by a variety of nontelomeric stress signals [7,8]. One well-established example of SIPS is oncogene-induced senescence (OIS), which was initially characterized when the Ras oncogene was overexpressed in primary mammalian cells [9] and was later associated with other oncogenes. Different treatment regimens, such as chemo-or radiotherapy, can also turn on SIPS, inducing a persistent DNA damage response (DDR) (genotoxic-induced) and a cancer cell proliferation blockade [10,11,12,13]. In addition, other types of senescence, such as epigenetically induced and mitochondrial dysfunction-induced senescence, have been described [14].

### 1.2. Characteristics of Senescent Cells

Senescent cells are commonly characterized by a range of features, regardless of the triggering signal. In addition to cell division arrest, these cells exhibit an aberrant shape, increased size, high granularity because of increased numbers of lysosomes and vacuoles, apoptotic resistance, a defective metabolism (higher glucose consumption and lactate production), and some chromatin alterations. Other molecular characteristics involve the expression of some tumour suppressor genes such as p16INK4A, ARF, p21 WAF1/Cip1, and p53; pRb dephosphorylation; senescence-associated heterochromatin foci (SAHF); increased lysosomal senescence-associated β-galactosidase (SA-β-gal) activity; and lipofuscin accumulation [15,16,17,18,19,20,21,22,23,24,25].

Arguably, the most remarkable attribute of senescent cells is the senescence-associated secretory phenotype (SASP), due to its effect on in the microenvironment and its important effects on a variety of disease pathologies. This phenotype involves the secretion of cytokines, chemokines, proteases, growth factors, extracellular media (ECM) elements and ECM-degrading enzymes. These molecules strongly affect the microenvironment via autocrine and paracrine senescence stimulation. On the one hand, the SASP can induce tissue repair through plasticity and stimulation of stemness and exert effects through the immune system to clear tumour cells. Several studies have suggested that cells that are subjected to transient SASP exposure exhibit increased expression of certain stem cell gene markers. The SASP is thought to evoke regenerative signals that induce cell plasticity and stemness, which is advantageous for tissue regeneration [26]. On the other hand, senescent cells can contribute to an immunomodulatory microenvironment and induce an inflammatory and catabolic state, which may eventually support carcinogenesis and metastasis, exhibiting undesirable effects of cellular senescence [26,27,28,29,30,31,32] (Figure 1).

## 2. Bimodal Role of Cellular Senescence

Substantial research has demonstrated that senescent cells can participate in advantageous and detrimental effects. Cellular senescence is fundamental for tissue homeostasis. Additionally, its activation during specific periods of embryonic and adult life is crucial for development, ensures clearance of stressed and damaged cells by immune cells, and exerts an antifibrotic function. In contrast, the persistent activation of cellular senescence and the accumulation of arrested cells in tissues and organs, due to the failed elimination of senescent cells from tissues, may be associated with aging, cancer and aging-related disorders [33,34] (Figure 1).

### 2.1. Senescent Cell Clearance by the Immune System

As mentioned above, senescent cells secrete different cytokines, chemokines and other molecules, called SAPS, that contribute to the clearance of senescent cells by immune cells, among other effects, but SASP is also related to some detrimental outcomes described previously. Previous studies have demonstrated that the accumulation of senescent cells accelerates with aging. In fact, the expression of certain senescence markers (such as p16^INK4a^) increases in aging tissues due to a higher number of senescent cells [35,36]. This phenomenon could result from an increase in the production of senescent cells or their insufficient clearance by immune cells [37]. Under physiological conditions, senescent cells are cleared by immune cells [38], but several authors have suggested that due to the decline in immune function in aged organisms, there is a decrease in the clearance of senescent cells [34]. However, there is evidence of persistent senescent cells in vivo. For instance, young adult women cured of breast cancer through treatment with cytotoxic chemotherapy exhibited increased expression of cellular senescence markers for decades [39]. Other studies suggest that commonly used chemotherapies can induce persistent senescent cells in noncancerous mouse tissue. In addition, research on cancer survivors showed that one long-term effect of chemotherapy is the accelerated development of a variety of age-associated diseases [40,41,42]. In conclusion, there is evidence of senescent cell clearance in some situations, but also examples of long-term senescent cell persistence in others.

### 2.2. Senescence and Cancer

Cancer is the result of the uncontrolled proliferation of cells, which can sometimes invade other tissues, causing metastasis. In cancer, senescent cells also have opposite effects. In the early stages, senescence acts as a tumour suppressor, decreasing cell proliferation in response to oncogene expression, preventing cancer cell proliferation and suppressing malignant progression. Additionally, senescent cells can attract immune cells to the tumor site, promoting the recognition and clearance of tumour cells by the immune system. However, in more advanced tumour stages, senescent cells may play a tumour-promoting role by driving carcinogenesis and metastasis through the SASP, inducing an immunosuppressive, proinflammatory and tumorigenic microenvironment. Indeed, in cancer patients treated with radiotherapies or chemotherapies, the induced senescence in tumour and healthy tissues has been related to the induction of stemness, relapses, metastasis, and a worse outcome [43]. Furthermore, different authors have recently demonstrated that senescent cells can enter the cell cycle in response to certain conditions and continue proliferating, acquiring aggressive behaviour. This cell plasticity is additional evidence of the deleterious outcomes of the dual effects of cellular senescence [13,34,44,45,46,47,48,49,50] (Figure 2).

An undesirable effect related to chemotherapy is accelerated aging and a higher incidence of age-related diseases in people with cancer [51], especially children. In fact, chemotherapies induce chronic cell senescence in healthy tissues [40,52,53].

## 3. Implications of Cellular Senescence in Diseases

The chronic presence of senescent cells has been associated with the onset of multiple diseases. In some preclinical studies, it has been shown that the depletion of senescent cells mitigates different chronic diseases such as frailty, cardiac dysfunction, vascular hyporeactivity and calcification, diabetes, liver steatosis, osteoporosis, intervertebral disc degeneration, pulmonary fibrosis, and radiation-induced damage [54,55]. Thus, the benefits of cellular senescence are not clear, and in fact, elimination of these cells may have benefits in many clinical settings. Therefore, it is important and useful to implement new therapies to target senescent cells to prevent their detrimental effects. In fact, there are different senotherapies under clinical and preclinical development; some of these therapies were originally identified as having antitumor effects.

## 4. Therapeutic Difficulties

Over the years, useful tools for specifically identifying senescent cells have rarely been available. Despite the current difficulties in the study of cellular senescence in vitro and in vivo, different approaches have been implemented in research on the molecular biology and behaviour of senescent cells.

β-Galactosidase activity is a distinctive feature of senescent cells (SA-β-gal), allowing researchers to identify senescent cells with a simple colorimetric assay. p16INK4A, a cyclin-dependent kinase inhibitor, is a crucial regulator in cellular senescence and a marker, as well as a regulator of ARF-mediated stabilization of the transcription factor p. To unequivocally identify senescent cells, researchers commonly use a combination of SA-β-gal, lipofuscin, loss of nuclear high-mobility group Box 1 (HMGB1) or lamin B1, increased levels of cell cycle inhibitors (i.e., p16INK4A) and p21 or SASP factors, such as cytokines, chemokines, proteases, growth factors, extracellular media (ECM) elements and ECM-degrading enzymes (Table 1) [9,20,21,27,32,40,56,57,58,59,60].

Once senescent cells were characterized, many studies were carried out to discover different drugs for eliminating or modulating their detrimental effects, a form of treatment known as senotherapy. However, many complications and technical challenges have been encountered. The behaviour of senescent cells in response to a particular stress stimulus varies depending on the cell lineage, since senescent cells derived from different cell types share common features but a distinctive molecular pattern and/or secretome with different SAPS factors. These differences make it difficult to study senotherapeutic drugs because senescent cells of different lineages show diverse responses when treated with a given compound. Furthermore, senescent cells differ depending on the origin of the senescence signal (oncogene activation, chemo- or radiotherapy, etc.), and their sensitivity to different senotherapies also differs. However, studies have focused on the similarities among senescent cells to target them [27,61]. Thus, for development of an effective senotherapeutic compound, some common rules are applied.

First, the treatment should match the precise senescent cell being targeted. Additionally, a causal relation between senescent cells, the disease and the mechanism by which senescent cells contribute to the illness (i.e., SAPS) should be well-defined. Indeed, due to the bimodal behaviour of cellular senescence, the risk-benefit balance should be evaluated in preclinical studies. Furthermore, conventional therapy for the disease might have important disadvantages (for instance, chemo- and radiotherapy side effects in non-tumoral tissues) that could be eliminated with the addition of or replacement by senotherapy. For this reason, senotherapy should have the least possible side effects on nonsenescent cells and a high impact on senescent cells. Indeed, a variety of molecules have been demonstrated to deplete senescent cells in vivo, but some of these compounds trigger undesirable secondary effects, such as thrombocytopenia [62]. Finally, senotherapeutic molecules should accumulate in the tissue where the senescent cell population or the SASP is being produced, or they should be administered continuously or in delayed-release formulations. Another aspect to consider is the long-term safety of the therapy because some have immunosuppressive effects [16].

On the basis of these considerations and the results of preclinical trials, osteoarthritis and atherosclerosis may be ideal candidates for senotherapy. Other age-related candidate illnesses include glaucoma and idiopathic pulmonary fibrosis. However, an important advantage of the osteoarthritis approach is that senotherapy could be administered locally, minimizing off-target effects [16]. Osteoarthritis is a degenerative joint disease characterized by the gradual loss of joint cartilage, distorted bone growth, joint inflammation and pain, which is common in the aged population [63,64]. Conventional therapy consists of nonsteroidal anti-inflammatory drugs [65]. The main cause has not been fully elucidated, but senescent cells have been reported to accumulate in osteoarthritis-affected joints. Nonetheless, the hypothesis that senescent cells have a causal role in the pathogenesis of this disease has been demonstrated only in mouse models of traumatic osteoarthritis. In this model, senescent chondrocytes accumulated at the articular surface of the arthritic joint, causing a deleterious effect on cartilage regeneration. The intra-articular administration of the senolytic compound UBX0101 (NCT04129944) reduced pain and promoted the repair of the affected cartilage, and a reduction in some SASP factors has been observed in some current clinical trials, see below [16,66].

## 5. Senotherapeutics

Senescent cells are an important research area due to their contributions to the pathogenesis of aged phenotypes, such as frailty, sarcopenia and aging-related disease (Figure 3).

Hence, a new pharmacological category of treatments, called senotherapeutics, is under development. This group includes senolytic drugs that selectively attack senescent cells and senostatic drugs that suppress SASP factor delivery, inhibiting senescent cell development.

RNA sequencing was used to examine the gene expression profile of nonsenescent and senescent preadipocytes induced by ionizing radiation. An increase in antiapoptotic regulators in senescent cells, such as PI3K/Akt pathway components and BCL-2 family members, confer resistance to apoptosis [67]. Thus, senescent cells were found to be protected from their own inflammatory SASP, allowing their survival, even though neighbouring cells died (Table 2). On the basis of these findings, the first senolytic drugs were developed and shown to inhibit some of these pathways and promote the apoptosis of the senescent cells dasatinib and quercetin.

### 5.1. First Senolytics: Dasatinib and Quercetin

Dasatinib is an oral dual BCR/ABL and Src family tyrosine kinase inhibitor [68]. This drug was approved for patients with chronic myelogenous leukaemia. Dasatinib promotes apoptosis, which depends on the activation of receptors such as BCRABL, SRC, ephrins and GFR [69,70]. Dasatinib induced the death of senescent human preadipocytes but was much less effective on senescent human umbilical vein epithelial cells (HUVECs) [67]. Quercetin is a natural flavonoid present in plants and an antioxidant similar to many other phenolic and heterocyclic compounds [71] that inhibit PI3K, other kinases, and serpines [72]. In this case, quercetin had a greater effect on senescent HUVECs than on senescent human preadipocytes. A reduction in cell viability was observed by ATPLite™ after three days of treatment with both drugs [67]. For this reason, the combination of these drugs induced the apoptosis of these two senescent types of cells (preadipocytes and HUVECs) more effectively than individual treatment and the drug did not act upon nonsenescent cells [67]. In addition, dasatinib alone diminished the p21 protein level, and the combination D+Q (dasatinib plus quercetin) reduced the levels of PAI-2 and BCL-xL. This combination treatment (D+Q) mitigated several senescence-associated phenotypes, proving the usefulness of senolytic drug applications in the improvement of the health span [67]. The first-in-human open-label clinical trial of senolytics provides initial evidence that D+Q significantly improved the physical dysfunction in subjects with idiopathic pulmonary fibrosis [73]. Similarly, another study demonstrated that a short course (3 days) of D+Q could decrease senescent cells in patients with diabetic kidney disease [74]. An ongoing clinical trial is testing the utility of this combination in the deceleration of aging and the risk of age-related diseases (NCT04946383) [74].

### 5.2. Inhibitors of BCL2

Inhibitors of BCL-2 family members, such as navitoclax or fisetin, can also act as senolytics. Navitoclax affects the viability of senescent but not proliferating HUVECs, IMR90 human lung fibroblasts and murine embryonic fibroblasts (MEFs), but not human primary preadipocytes. The targets of this drug are Bcl-2, Bcl-xl, and Bcl-w [75]. Currently, navitoclax is being investigated to treat myelofibrosis, and phase 3 clinical trials are ongoing (NCT04472598, NCT04468984). A recent study to test the response to aging-induced neurovascular coupling, also named functional hyperaemia (NVC), showed that treatment with navitoclax improved this condition in aged mice [76]. In addition, the administration of navitoclax to either sublethally irradiated or normally aged mice effectively killed senescent cells (senescent bone marrow haematopoietic stem cells and senescent muscle stem cells) [41]. This drug has haematological toxicity due to its lower specificity [75]. The recovery in the selectivity of the drugs reduced the haemotological side effects caused by navitoclax. For this reason, A1331852 and A115463, which are selective inhibitors of BCL-XL, may be better candidates for translation into clinical applications. These two inhibitors act on senescent HUVECs and IMR-90 cells but not primary human preadipocytes [77].

The next inhibitor, fisetin, is a flavonoid that partially inhibits BCL-2 family members such as BCL-xL, HIF-1a and other SCAP network components [78]. This molecule activates caspase 7, 8 and 9, promoting apoptosis of MCF-7 human breast cancer cells but not nontumorigenic cells [79]. Fisetin selectively induces apoptosis in senescent but not proliferating HUVECs. Unlike navitoclax, fisetin does not act as a senolytic agent in senescent IMR90 cells, a human lung fibroblast strain, or primary human preadipocytes [75,77,78,79]. There are some ongoing clinical trials in phase 2 for fisetin. One of them is studying the clinical efficacy of this drug to attenuate osteoarthritis-related articular cartilage degeneration (NCT04210986). Another clinical trial is focused on frailty in adult survivors of childhood cancer. In this case, they will employ two different treatment regimens, D+Q and fisetin (NCT04733534), to evaluate and compare the efficacy, safety, and tolerability of D+Q and fisetin in the reduction of cellular senescence and the improvement of frailty. In addition, some clinical trials with COVID-19 patients used fisetin to reduce an excessive inflammatory reaction, which is one of the complications of SARS-CoV-2 infection (NCT04476953; NCT04537299, NCT04771611) [73,74] (Table 2).

### 5.3. Piperlongumine

Piperlongumine is another natural alkaloid that has senolytic activity. This compound is found in trees of the *Piper* genus [80]. It preferentially acts on senescent WI-38 fibroblasts induced by IR exposure, ectopic expression of the oncogene Ras or replication exhaustion over nonsenescent WI-38 cells. However, the mechanism of its senolytic activity is not yet clear. Piperlongumine could promote apoptosis and suppress tumour growth in vitro and in vivo by inhibiting the DNA binding activity of nuclear factor-κB (NF-κB) in lung cancer due to a direct interaction between piperlongumine and the p50 subunit of NF-κB [81].

### 5.4. Inhibitors of HSP90

Another group consists of the inhibitors of the molecular chaperones heat shock protein 90 (HSP90), geldanamycin and tanespimycin (17-AAG) [80,81], which protect cellular proteins from degradation. Although research on the role of HSP90 in senescence is unclear and incomplete, the repression of senescence by HSP90 is caused by the degradation of p14ARF, TERT stabilization and SASP induction and the increase in senescence due to the upregulation of AKT expression and maintenance of the DDR response [82]. Geldanamycin is a natural antitumor antibiotic, and its semisynthetic analogue 17-AAG showed lower toxicity [83,84,85]. As an anticancer drug, 17-AAG has undergone clinical trials, but it was deemed unsuccessful as a direct antitumor compound [86]. However, these molecules may act through the downregulation of the PI3K/Akt signalling pathway [87]. Cellular stress and heat shock cause the activation of full-length HIV transcription-producing infectious virus from dormant HIV genomes [88]. During heat shock, the molecular chaperone heat shock protein 90 (Hsp90) is overexpressed. An inhibitor of Hsp90, 17-AAG, could prevent expression of the HIV gene in replication-competent cellular reservoirs that would typically cause a rebound in plasma viremia after antiretroviral therapy (ART) cessation [88]. This compound would be an adjuvant to current ART regimens to suppress rebound viremia from persistent HIV reservoirs.

### 5.5. Panobinostat

A recent study suggested the potential of a nonselective histone deacetylase inhibitor called panobinostat as a senolytic drug [89]. Panobinostat is a class I, II and IV HDAC inhibitor (HDACi) that was FDA-approved for the treatment of refractory multiple myeloma. The combination of this drug and Taxol had a synergistic effect on panobinostat senolytic action. Panobinostat could induce a G2 block [90], which explains the antagonism observed in the combination treatment [89]. Single and combination treatments increased the mRNA expression of SASP-related factors and decreased Cyclin A2 expression. With a combination of these drugs, the cells showed a flattened, granular morphology coupled with SA-β-gal positive staining. After chemotherapy, cell populations exhibited a decrease in H3 acetylation, which could be reversed by panobinostat. Thus, post-treatment with a histone deacetylase inhibitor was shown to be a more efficacious therapeutic approach than repeated doses of treatment standards, such as Taxol or cisplatin [89]. Consequently, panobinostat may be used as a drug targeted to persistent senescent cells after treatment with conventional drugs.

### 5.6. Cardiac Glycosides

Senescent cells present a small amount of plasma membrane depolarization and higher level of H+. Through the Na+/K+ATPase pump, cardiac glycosides (Digoxin and Ouabain) induce a dysregulation in the electrochemical gradient inside the cells causing depolarization and acidification [91]. Furthermore, treatment with ouabain or digoxin promoted apoptosis in senescent cells, which was caused in part by the increase in the levels of some proapoptotic BCL2 family proteins and NOXA [92]. Hence, senescent cells could be more susceptible to the action of these agents. The senolytic activity was tested in vitro and in vivo, and it was found to be effective against tumour and primary cells [91,92]. However, cardiac glycosides could be employed as anticancer drugs because they induce immunogenic cell death, and recent studies have shown a reduction in atherosclerosis [93] and bleomycin-induced pulmonary fibrosis [94].

### 5.7. FOXO4-TP53 Disrupting Peptides

FOXO4 is a transcription factor that plays a role in the maintenance of senescent cell viability. In addition, its presence in a small fraction of nonsenescent adult cells makes it a potential target to eliminate senescent cells. FOXO4 binds p53 and prevents it from inducing the apoptosis of senescent cells. For this reason, impairing this FOXO4-TP53 interaction may be useful to activate apoptosis in these cells. FOXO4-DRI is a cell-permeable peptide that includes part of the p53-interaction domain of FOXO. This peptide could compete with endogenous FOXO4 for p53, impairing the FOXO4-p53 interaction, and inducing apoptosis in senescent cells. Several studies have shown that FOXO4-DRI selectively eliminates senescent cells, without affecting nonsenescent ones. Another peptide recently developed by molecular modelling is ES2, which has the same function as the previously described [95,96,97]. The authors demonstrated that combination therapy of ES2 and a Braf inhibitor results in apoptosis and a survival advantage in mouse models of Braf mutant melanoma and reduced senescent cells in ageing mice. Furthermore, they showed that ES2 is effective at eliminating both normal and cancer senescent cells [98].

### 5.8. Other Senolytics

Fenofibrate (FN), a PPARα agonist used for dyslipidaemias in humans, was identified as a senolytic. This agent induced the apoptosis of senescent cells, increased autophagy and protected against cartilage degradation. Fibrate treatment improved osteoarthritis in patients from the Osteoarthritis Initiative (OAI) cohort in a retrospective study [99].UBX0101 is a p53/MDM2 interaction inhibitor. However, the mechanism by which senescent cell apoptosis is induced has not been fully elucidated. Intra-articular injection of UBX0101 selectively caused the clearance of senescent cells that are accumulated in the articular cartilage. More beneficial effects were the reduction in the development of post-traumatic osteoarthritis and the increase in the chondrogenesis [66]. Some clinical trials are investigating this inhibitor in knee arthritis (NCT04129944, NCT04349956, NCT04229225, NCT03513016).

As previously mentioned, another group of drugs called senostatics indirectly inhibits senescent cells by attenuating SASP compound activity. This group of drugs includes rapamycin, ruxolitinib, metformin and resveratrol (Table 2).

### 5.9. Inhibitors of mTOR Kinase

Rapamycin is a natural macrolide that acts as a selective inhibitor of mTOR kinase in the mTORC1 complex, a key molecule involved in cell proliferation and survival [100]. It has been used in the clinic as an immunosuppressant [101]. Rapamycin caused the reduction of senescence through a nuclear factor E2-related factor 2 Nrf2-independent mechanism and was correlated with suppression of SASP [102]. This activity against SASP was correlated with the downregulation of interleukin (IL)-6 expression and inhibition of IL-1α translation [103]. Two studies showed that the activity of rapamycin inhibited senescence but did not eliminate senescent cells [102,103]. Currently, there is a clinical trial in recruitment to study the long-term safety profile and the efficacy of rapamycin in reducing clinical measures of aging in an older adult population (NCT04488601). Side effects have been described with chronic administration and includes the induction of ulceration of the mucosal tissues, haematological abnormalities, insulin insensitivity, obesity, and diabetes, although these adverse effects may be largely dose dependent. Everolimus, an analogue of rapamycin, recruits the immunophilin/proly isomerase FKBP12 to mTORC1 and has better bioavailability and pharmacokinetics [104]. The second generation of mTOR inhibitors competes with ATP for the active site of mTOR kinase inhibiting both mTORC1 and mTORCAZD8055 has high specificity and selectivity for mTOR kinase. ATP-competitive inhibitors show more apoptotic effects in vitro than rapamycin or everolimus, but in vivo, they have not yet demonstrated better efficacy than current treatment regimens [105].

### 5.10. JAK Inhibitors

The JAK/STAT pathway has an important role in chronic inflammation related to aging and age-related diseases [106]. For this reason, inhibitors of JAK, such as ruxolitinib, reduced the SASP in irradiation-induced senescent cells and systemic and adipose tissue inflammation in frail mice [107]. In addition, the premature aging phenotype was delayed by this drug in a murine model of progeria [108]. A recent clinical trial (NCT02475655) verified the good tolerability of ruxolitinib in HIV-infected persons and the significantly decreased markers of immune activation and cell survival (HLA-DR/CD38, CD25, CD127, Bcl-2, Ki-67, α4β7) [109].

### 5.11. Metformin

Metformin, an antidiabetic drug, was discovered to have a possible effect on senescence because of its beneficial effects on aging and lifespan [110]. Metformin inhibited the phosphorylation of IKb and IKKa/b stimulated by lipopolysaccharide (LPS) in *ampk* null fibroblasts and in macrophages, resulting in inactivation of the NFKβ pathway [111]. Regarding lifespan, metformin increased the lifespan and delayed tumour development when treatment was started at young and middle ages but not during old age in female SHR mice [110,111,112,113,114]. Clinical trials including the MILES (Metformin In Longevity Study) and TAME (Targeting Aging with Metformin) have been designed to test the beneficial effects of this agent as an antiaging agent. The main issue is whether healthy people will show this protective effect of metformin. Some authors attributed the increase in lifespan to its effects on cellular metabolism [115].

### 5.12. Resveratrol

Another drug that has demonstrated innumerable biological activities is resveratrol [116,117,118]. This flavonoid is present in wine and fruits and is beneficial for cancer chemoprevention, provides protection against cardiovascular diseases, and has anti-inflammatory activity and corrective effects on metabolism [119,120]. Resveratrol had beneficial effects on aging and attenuated the development of the SASP in human MRC5 fibroblasts through a reduction in the release of proinflammatory cytokines without producing changes in senescence [121,122]. Furthermore, this drug induces apoptosis via the generation of ROS in diffuse large B-cell lymphoma cell lines and inactivated the AKT/PKB pathway. Resveratrol has also been reported to increase the lifespans of nematodes and yeast [116,117,118] and to prevent age-related diseases in elderly humans, but not in mice [119,120]. Nevertheless, there are some studies which suggest that the lack of significant findings regarding the beneficial metabolic effects of this agent in nonobese post-menopausal women [123] and healthy obese individuals [124]. The protective effects against atherosclerosis are only observed in individuals without high risk [125].

### 5.13. ATM Inhibitor

The ataxia-telangiectasia mutated (ATM) kinase inhibitor, KU-60019, was described as a senostatic. The activity of this drug caused functional recovery of the lysosome/autophagy system, mitochondrial function and metabolic reprogramming. Therefore, senescence could have other regulatory mechanisms, such as the lysosomal-mitochondrial axis modulated by ATM activity [126] (Table 2).


cells-11-01222-t002_Table 2Table 2Overview of senotherapeutic.
AgentMechanism of ActionClinical StatusClinical IndicationClinical TrialEffectSide EffectsDoses
**Senolytic**

**Dasatinib**
(-) Src family tyrosine kinaseApproved to (1)Experimental to (2) and (3) (phase 2)(1) Chronic myelogenous leukemia(2) Epigenetic aging(3) FrailtyNCT04946383NCT04733534↑ Lifespan↓ Osteoporosis  AtherosclerosisBlood countsDiarrheaBleedingFever100–140 mg/day orally
**Quercetin**
PI3K antagonistExperimental (phase 2)Epigenetic agingFrailtyNCT04946383NCT04733534NauseaKidney damage420–1400 mg/m^2^ intravenous (IV) bolus once/week
**Navitoclax (ABT-263)**
BCL-2 antagonistExperimental (Phase 3)Myelofibrosis[65]NCT04472598NCT04468984↑ Hematopoiteticstem cell function↓ Atherosclerosis
Diarrhea NauseaThrombocytopenia Lymphocitopenia150 mg/7-day/325 mg on continuous (21/21) schedule (Clin trial)
**Fisetin**
(-) BCL-xL, HIF-1α and other SCAP network componentsExperimental (Phase 2)OsteoarthritisFrailtyCOVID-19NCT04210986NCT04733534NCT04476953NCT04537299NCT04771611Not describedNo evidence in animal studies20 mg/kg for two consecutive days (Clinical trial)
**Piperlongumine**
GSTP1 antagonistExperimental-[70]Not describedNot describedNot described
**Geldanamycin**
(-) HSP90Experimental-[73,74,75]↑ Lifespan↓ Age related symptomsNot describedNot described
**Tanespimycin (17-AAG)**
ExperimentalSolid tumorsMultiple myeloma[76]DiarrheaNauseaHepatotoxicityAnemiaThrombocytopenia56–450mg/m^2^ on different dosing schedules (no established)
**Panobinostat**
(-) nonselective histone deacetylaseApproved to (1)(1) Multiple myeloma[77]Not describedDiarrheaSevere and fatal cardiac events20 mg/3 doses/week in combination with Bortezomib and dexamethasone
**Senostatic**

**Rapamycin**
(-) mTOR kinaseApproved to (1)(1) Immunesuppression[80,81]NCT04488601↑ LifespanHeadacheDiarrheaNauseaJoint pain16–24 mg/µL for prophylaxis of Renal Transplant Rejection
**Ruxolitinib**
(-) JAKApproved to (1)(1) Myeloproliferative diseases[82]Not describedAnemiaThrombocytopeniaBruisingNeutropenia 5–20 mg orally twice daily
**Metformin**
AMPK agonist(-) mGPDApproved to (1)(1) Diabetes type II[84,85]↑ Lifespan in miceLactic AcidosisVitamin B12 deficiencyHypoglycemia500–2000 mg orally once daily
**Resveratrol**
(-) Histone deacetylaseExperimental -[91,92]↑ Lifespan in *cerevisae*No evidence250-1000 mg daily for up to 3 months.


## 6. Senotherapeutics for HIV Infection

The life expectancy of persons living with HIV (PLWH) on ART has improved worldwide in recent years [127]. However, despite effective ART and virologic control, it is well known that HIV causes a persistent low-grade inflammatory and immune activation state [128,129,130,131]. Inflammation is a key factor in physiological aging (inflammaging) [132]. For this reason, PLWH undergo accentuated and accelerated aging [133,134] that increases the risk of aging-related diseases [135,136]. Because inflammaging is part of the altered immune response associated with aging, inflammatory biomarkers (IL-1, IL-6, TNF) could be considered as biomarkers of interest [137].

On a mechanistic level, this accelerated aging is thought to occur due to increased immune activation, chronic inflammation, oxidative stress and mitochondrial dysfunction maintained even when HIV viral replication has been suppressed for years [96,97,98,99]. The causes are multiple and are due, in a direct and indirect way, to HIV infection [131]. Some HIV proteins (i.e., Nef) induce the secretion of proinflammatory cytokines and chemokines [138]. In the same way, coinfection with other viruses (i.e., cytomegalovirus) has been associated with inflammation and immune dysfunction [139]. Another reason could be the dysfunction of the intestinal barrier that allows bacterial translocation [140], promoting inflammation and immunoactivation (Figure 4).

In vivo and in vitro evidence has shown that HIV contributes to cellular senescence [141,142,143,144]. Therefore, in contrast to healthy controls, untreated PLWH have increased p16 INK4a levels [145]. Excessive accumulation of senescent cells in numerous tissues leads to multiple chronic diseases, tissue dysfunction, age-related diseases and organ aging [136]. Indeed, some studies have suggested that long-term ART might influence immune senescence in people living with HIV [145,146,147,148].

To date, different therapeutic strategies have been explored in PLWH to reduce immune activation and inflammation, but none of them have been fully effective [131]. These strategies include early ART initiation, ART intensification, the use of anti-inflammatory agents, the rebuilding the gut microbiota system, and so on. Because senotherapeutics have improved the health span and lifespan in mouse models of disease and in early trials with human patients, they may be an interesting alternative for treating HIV immune cell senescence [149]. In fact, senotherapy could ameliorate these disorders by targeting senescent cells or their SASP and open the door to new treatment options against aging and chronic diseases [128,129,130,135,141,142,143,144,145,146,147,148,150,151,152,153,154].

One of the main limitations of curing HIV is the ineffectiveness of ART in reducing the size of the HIV-1 latent reservoir. Recent studies have demonstrated the association of the BCL-2 protein with cells that support proviral forms in the setting of latency [155]. In the same way, previous studies have observed that the HIV Tat protein increases BCL-2 expression in monocytes, which would favour a mechanism of HIV-persistent infection in monocytes [156]. Therefore, blocking BCL-2 could be a promising therapeutic strategy for curing HIV. To date, there are several ongoing clinical trials (www.clinicaltrials.gov accessed on 22 February 2022) on the use of senotherapeutics in PLWH (see for example NCT02440789, NCT02429869, NCT02475655).

In conclusion, although the correlation between HIV infection and accelerated aging is consistent, the association with cellular senescence and the mechanisms contributing to it are not fully understood. For this reason, additional extensive studies are necessary to shed light on these mechanisms. Viral persistence and the chronic immune activation and exhaustion that characterize treated patients with HIV suggest that senolytic and senomorphic strategies may be effective and confer advantages to these patients [149,157] (Table 3).

## 7. Conclusions and Future Perspectives

Our understanding of senescence and its involvement in different diseases has increased in recent years. Nevertheless, it is necessary to shed light on the specific role of senescent cells in every single pathology and the potential of therapeutically targeting senescent cells to improve the standard treatment in these illnesses.

In this review, we focus on the physiology and characteristics of senescence and its implications in cancer and HIV pathology, among other pathologies. Furthermore, we summarize the most recent advances in the development of a new category of drugs that target senescent cells or their secreted molecules. Senotherapeutics involve a new group of treatments that directly attack senescent cells (senolytics) or inhibit SASP factor delivery by senescent cells (senomorphics). These molecules have demonstrated senescence-modulating properties, and we describe their experimental evaluation as promising candidates for the treatment of a variety of diseases in which senescent cells have been studied. However, progress in the discovery, validation and development of new active molecules is poor. Indeed, almost all currently validated senotherapeutics are drugs that are currently being used for the treatment of well-known diseases, which have recently demonstrated their therapeutic potential against senescent cells. Furthermore, the signalling pathways or specific targets that direct these senotherapeutics against senescent cells are underexplored. For these reasons, research efforts need to focus on the advancement of the search for new targets in senescent cells to improve this landscape of senotherapeutics, which is still in its infancy.

In conclusion, targeting senescence could be potentially beneficial for improving the healthspan in a variety of illnesses, such as age-related diseases, frailty, fibrosis, cancer or HIV persistence, and their clinical consequences. Nevertheless, the development and applications of senotherapeutics are beginning to emerge, although critical aspects are far from being solved.

## Figures and Tables

**Figure 1 cells-11-01222-f001:**
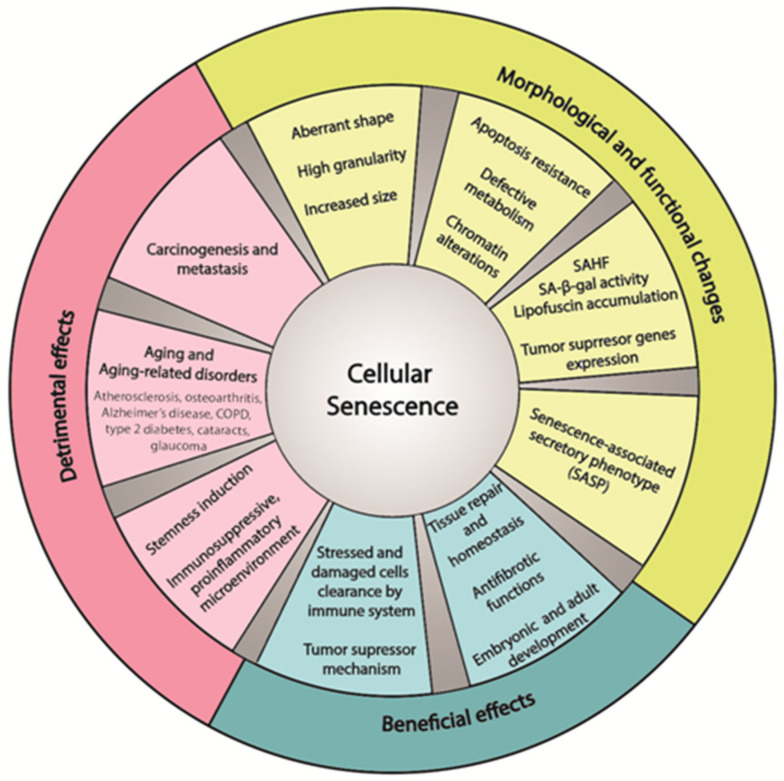
Characteristics, advantages and disadvantages of cellular senescence. Note: COPD: chronic obstructive pulmonary disease; SAHF: senescence-associated heterochromatin foci; SASP: Senescence-associated secretory phenotype.

**Figure 2 cells-11-01222-f002:**
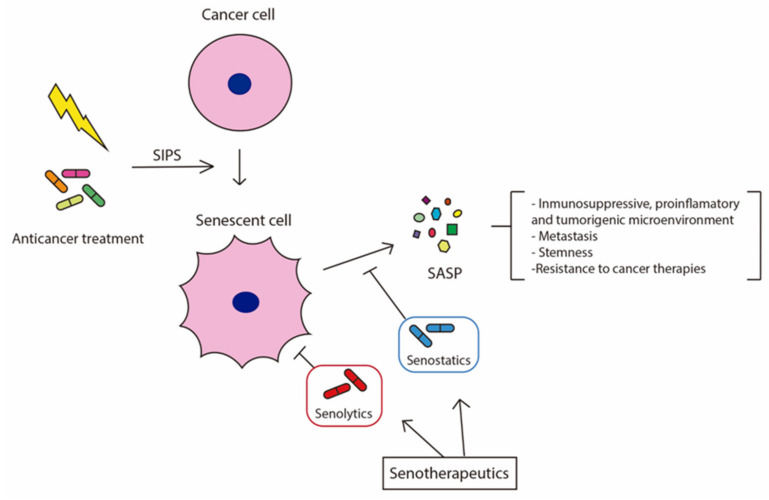
Induction of senescence by the anticancer treatment of tumor cells and the effects of senescent cells and secreted SASP in cancer. Senotherapies in cancer try to remove chemotherapy-induced senescent cells and/or block deleterious SASP.

**Figure 3 cells-11-01222-f003:**
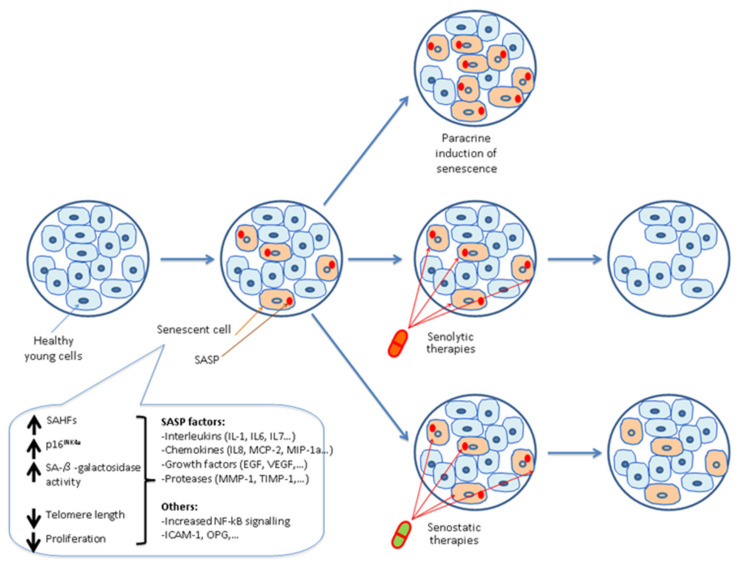
Scheme of senotherapeutics as a new pharmacological category of treatments.

**Figure 4 cells-11-01222-f004:**
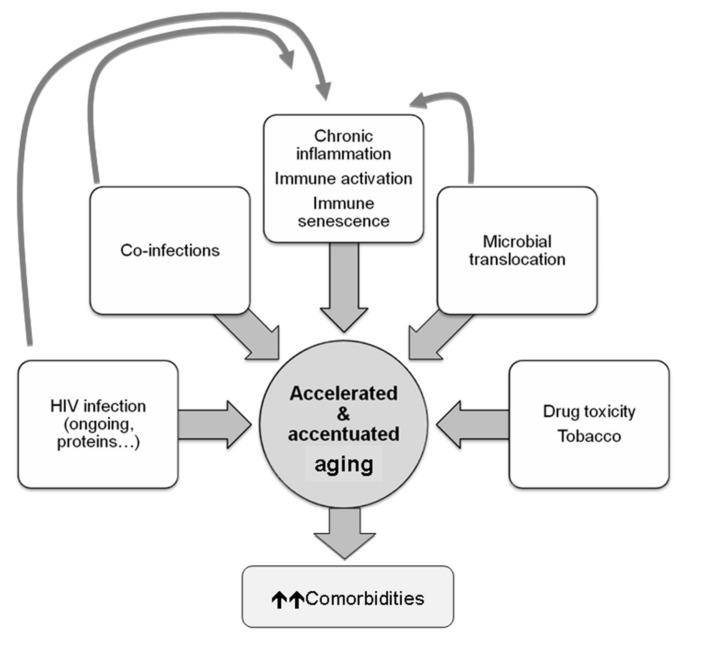
Main causes and consequences of HIV aging.

**Table 1 cells-11-01222-t001:** SASP factors.

SASP Factors	
**Soluble factors**	
**Interleukins (IL):** IL-6, IL-7, IL-8, IL-10, IL-1a, -1b, IL-13, IL-15	**Soluble or shed receptors or ligands** ICAM-1, -3, OPG, sTNFRI, TRAIL-R3, Fas, sTNFRII, Fas, uPAR, SGP130, EGF-R
**Chemokines (CXCL, CCL):** CCL2, CCL20, GRO-a,-b,-g, MCP-2, MCP-4, MIP-1ª, MIP-3ª, HCC-4, Eotaxin, Eotaxin-3, TECK, ENA-78, I-309, I-TAC, CXCL1, -2, -5, -11, -12.	**Nonprotein soluble factors** PGE2, Nitric oxide, Reactive oxygen species
**Other inflammatory factors:** GM-CSE, G-CSE, IFN-γ, BLC, MIF	**Insoluble factors (ECM)** Fibronectin, Collagens, Laminin
**Growth factors and regulators:** Amphiregulin, Epiregulin, Heregulin, EGF, bFGF, HGF, KGF (FGF7), VEGF, Angiogenin, SCF, SDF-1, PIGF, NGF, IGFBP-2, -3, -4, -6, -7, GDNF, PDGF,	
**Proteases and regulators:** MMP-1, -3, -9, -10, -12, -13, -14, TIMP-1, TIMP-2, PAI-1, -2; tPA; uPA, Cathepsin B	

**Table 3 cells-11-01222-t003:** Some of the main senolytics tested in HIV.

Agent	
Quercetin	Senolytic Reactivation of latent HIV-1 gene expression [158] PI3K antagonist
Venetoclax (ABT-199)	Senolytic BCL-2 antagonist Apoptosis of tumor cells Treatment of chronic lymphocytic leukemia [159] Selective killing of HIV-infected cells, resulting in decreased numbers of HIV DNA-containing cells [160]
Rapamycin	Senomorphic mTOR inhibitor Could diminish HIV reservoir expansion, persistence, and resistance to immune surveillance [72] Potential affection of the level of HIV persistence during effect therapy [161]
Ruxolitinib	JAK1/JAK2 inhibitor Senomorphic Inhibitor of HIV-1 replication and virus reactivation [162]
Dasatinib	Senolytic Broadly active TKI Diabetic kidney disease [74] Inhibition of s HIV-1 replication through the interference of SAMHD1 phosphorylation in CD4+ T cells [163]
Tanespimycin (17-AAG)	Senolytic HSP90 inhibitor Inhibition of Hsp90 activity prevents HIV gene expression in replication-competent cellular reservoirs [88]
Panobinostat	Senolytic HDAC inhibitor Significant reductions in multiple established plasma markers of inflammation; significant reduction in the proportions of intermediate monocytes and tissue factor-positive monocytes [164]
Everolimus	Senomorphic mTOR inhibitor Could diminish HIV reservoir expansion, persistence, and resistance to immune surveillance [72]
Baricitinib	Can reduce the HIV reservoir in lymphoid tissue derived cell [165] Reduction of the HIV-induced neuroinflammation marked by glial activation [166]
Metformin	Senolytic/Senomorphic AMPK agonist and glycerophosphate dehydrogenase (mGPD) inhibitor Significant decrease on CD4^+^ T-cell infiltration in the colon, significant decrease on mTOR activation/phosphorylation, especially in CD4^+^ T-cells expressing the Th17 marker CCR6, decrease of the HIV-RNA/HIV-DNA ratio [167]

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
