# Peer review of "Senotherapeutics in Cancer and HIV"

_cells, 2022, doi:10.3390/cells11071222_

Round 1

Reviewer 1 Report

The manuscript titled “Senotherapeutics in Cancer and HIV” is very confusing because of the following reason:

  1. The subheading 1.1. under heading 1 is missing.
  2. It seems the authors want to describe different types of senescence inducers in section 1. In
  3. Under heading 2 “Bimodal role of senescence”, there should be following two subheadings:                                                          Advantages of senescence.                                                     Disadvantages of senescence

Senescent cells claearance by the immune system is confusing under this heading. And then suddenly senescence and cancer.

  1. The first line of section 3 is copy and pasted from the following:

https://www.nature.com/articles/s41580-020-00314-w

It is written we highlight atherosclerosis, osteoarthritis,……..I think many of them are not highlighted here. The article is about cancer and HIV.

  1. Under heading senothreapeutics, line 269-281, role of Hsp90 inhibitors in senescence is described. In line 271, it is written “17-AAG is used”. It is not used in the treatment of cancer.

A recent paper titled “Recent update on discovery and development of Hsp90 inhibitors as senolytic agents”  (https://www.sciencedirect.com/science/article/pii/S0141813020335601)

Should be referred.

  1. In the conclusions, line 395 and 396, it is written that “we focus on the physiology and characteristics of senescence and its implications in cancer and other pathologies”. However, the heading says it is about senescence in cancer and HIV.

Other comments.

In vitro and in vivo should be in italics.

The English needs to be improved. Like in line 99 and 100. It is difficult for me to highlight so many mistakes.

Overall the manuscript is not suitable for publication in its current form.

It should be revised and written fresh with many figures and tables and thereafter submitted in a completely new form.

Author Response

The manuscript titled “Senotherapeutics in Cancer and HIV” is very confusing because of the following reason:

  1. The subheading 1.1. under heading 1 is missing. Corrected
  2. It seems the authors want to describe different types of senescence inducers in section 1. This part has been improved explaining the different types of senescence.
  3. Under heading 2 “Bimodal role of senescence”, there should be following two subheadings:     Advantages of senescence.    Disadvantages of senescence

Senescent cells claearance by the immune system is confusing under this heading. And then suddenly senescence and cancer.

We didn’t intend to remark the senescent cells clearance by the inmune system as an advantageous effect of senescence, but as an example of the bimodal role of cellular senescence. In this section, we explain that senescence can induce a clearance by the immune system of cells that are aged, damaged or stressed through SASP, but also SASP could bring disadvantageous effects (as oncogenic effects, inflammation, etc.). In the following section (2.2. Senescence and cancer), we address another example where senescence could act as a beneficial phenomenon, but also as a detrimental one in different situations.

  1. The first line of section 3 is copy and pasted from the following: https://www.nature.com/articles/s41580-020-00314-w

It is written we highlight atherosclerosis, osteoarthritis,……..I think many of them are not highlighted here. The article is about cancer and HIV.

The sentence has been corrected

  1. Under heading senothreapeutics, line 269-281, role of Hsp90 inhibitors in senescence is described. In line 271, it is written “17-AAG is used”. It is not used in the treatment of cancer. A recent paper titled “Recent update on discovery and development of Hsp90 inhibitors as senolytic agents”  (https://www.sciencedirect.com/science/article/pii/S0141813020335601) Should be referred.

Corrected

“Although the research of the role of HSP90 in senescence is unclear and uncompleted, the repression of senescence by HSP90 is caused through the degradation p14ARF, TERT sta-bilization and SASP induction and the increase of senescence due to the upregulation of AKT and maintenance of DDR response.”

“Geldanamycin is a natural antitumor antibiotic, and its semisynthetic analog 17-AAG improved due to its lower toxicity.”

In the conclusions, line 395 and 396, it is written that “we focus on the physiology and characteristics of senescence and its implications in cancer and other pathologies”. However, the heading says it is about senescence in cancer and HIV.

“In this review, we focus on the physiology and characteristics of senescence and its implications in cancer and HIV pathologies.”

Other comments.

In vitro and in vivo should be in italics.

It has been corrected

The English needs to be improved. Like in line 99 and 100. It is difficult for me to highlight so many mistakes.

The manuscript has been edited by native English spoken editors at AJE.

“For instance, young adult women cured of breast cancer through treatment with cyto-toxic chemotherapy, exhibited increased expression of cellular senescence markers for decades, among other examples. In conclusion, there is evidence of senescent cells clearance in some situations, but also examples of long-term senescent cells persistence in others.”

Reviewer 2 Report

This manuscript by Sánchez-Díaz et al. reviews a critically important subject of increasing interest to scientists across a steadily broadening area of research.  As such, I believe it is a timely submission that should be published and will attract substantial readership.  Overall, positive and negative aspects of cellular senescence are comprehensively reviewed in the text and concisely summarized in figure 1.  Table 1 summarizes candidate senolytic and senostatic agents with useful references and information on efficacy and side effects.  All figures are clear and the text is generally well written and easy to understand.  I have a few minor comments that could improve the quality of the manuscript from my perspective, should the authors decide to address them.

  1. Has there been any follow-up to experiments with the FOXO4-p53 peptide used to induce apoptosis of senescent cells?  https://doi.org/jcell.2017.03.031
  2. A little more information explaining the concept and/or mechanism of stemness induction in relation to the SASP would be helpful.
  3. Several of the candidate senolytic agents are alkaloids and another is a flavonoid. Is there a potential common mode of action attributable to their chemistry?
  4. I believe a number of studies with metformin and with resveratrol have shown no effect on certain age-related abnormalities. Some reference to these and to potential collateral effects of powerful drugs such as rapamycin, kinase inhibitors and histone deacetylase inhibitors would be good for balance.
  5. In the abstract, the authors refer to HIV as a disease and refer to it as an event in the introduction. Neither is strictly correct with “HIV infection” contributing to physiological abnormalities that involve cellular senescence.
  6. Much more could be made of the role of cytomegalovirus, contributing to immune senescence in its own right, and accelerating aging in HIV infection both through driving inflammation and driving immune senescence.
  7. Line 45 appears grammatically incorrect (“inducers that mechanisms…”)
  8. Line 67 is grammatically incorrect. (“although” should be “through”?)
  9. On line 81, the abbreviation SAPS is used. I may have missed it, but couldn’t find the definition.
  10. The meaning of lines 275-276 is unclear. (“in” should be “from”?)

Author Response

Comments and Suggestions for Authors

This manuscript by Sánchez-Díaz et al. reviews a critically important subject of increasing interest to scientists across a steadily broadening area of research.  As such, I believe it is a timely submission that should be published and will attract substantial readership.  Overall, positive and negative aspects of cellular senescence are comprehensively reviewed in the text and concisely summarized in figure 1.  Table 1 summarizes candidate senolytic and senostatic agents with useful references and information on efficacy and side effects.  All figures are clear and the text is generally well written and easy to understand.  I have a few minor comments that could improve the quality of the manuscript from my perspective, should the authors decide to address them.

  1. Has there been any follow-up to experiments with the FOXO4-p53 peptide used to induce apoptosis of senescent cells?  https://doi.org/jcell.2017.03.031

“5.7    FOXO4-TP53 disrupting peptides

FOXO4 is a transcription factor which plays a role in the maintenance of senescent cell viability. In addition, its presence in a small fraction of non-senescent adult cells, makes it a feasible target to eliminate senescent cells. FOXO4 binds p53 and prevents it to induce apoptosis of senescent cells. For this reason, impairing this FOXO4-TP53 interac-tion may be useful to activate apoptosis in these cells. FOXO4-DRI is a cell-permeable pep-tide that includes part of the p53-interaction domain of FOXO4. This peptide could com-pete with endogenous FOXO4 for p53, impairing the FOXO4-p53 interaction, and induc-ing apoptosis in senescent cells. Several studies showed that FOXO4-DRI selectively eliminate senescent cells, without affecting non-senescent ones. Another peptide recently developed by molecular modelling is ES2, with the same function as the previously de-scribed. The authors demonstrated that the combination therapy of ES2 and a Braf inhibitor results in apoptosis and a survival advantage in mouse models of Braf mutant melanoma and reduced senescent cells in ageing mice. Furthermore, they suggest that ES2 is effective at eliminating both normal and cancer senescent cells.”

  1. A little more information explaining the concept and/or mechanism of stemness induction in relation to the SASP would be helpful.

“Several studies suggest that cells that are brought under a transient SASP exposure, increased certain stem cell genes markers expression. SASP is thought to evoke regenerative signals that induce cell plasticity and stemness, which is advantageous for tissue regeneration.”

  1. Several of the candidate senolytic agents are alkaloids and another is a flavonoid. Is there a potential common mode of action attributable to their chemistry?

In order to flavonoids, it was proposed a possible mechanism through which this drugs could have effect as senolytic. Flavonoids show a potent prooxidant activity and this effect is increased in presence of transition metals such as copper and iron. Senescence cells presents high levels of these metals. So that, quercetin and fisetin could have a selective mechanism associated with metals-promoted oxidative damage in senescence cells.

Reference: Wang Y, He Y, Rayman MP, Zhang J. Prospective Selective Mechanism of Emerging Senolytic Agents Derived from Flavonoids. J Agric Food Chem. 2021 Oct 27;69(42):12418-12423. doi: 10.1021/acs.jafc.1c04379. Epub 2021 Oct 18. PMID: 34662116

  1. I believe a number of studies with metformin and with resveratrol have shown no effect on certain age-related abnormalities. Some reference to these and to potential collateral effects of powerful drugs such as rapamycin, kinase inhibitors and histone deacetylase inhibitors would be good for balance.

We added the studies found to resveratrol in which there aren’t activity in age-related disorders. Moreover, the side effects of drugs appear in table 1, rapamycin, among others.

In the case of metformin:

“Clinical trials including the MILES (Metformin In Longevity Study) and TAME (Targeting Aging with Metformin), have been designed to test the beneficious effects of this agent as an anti-aging. The mainly doubt is if healthy subjects would have this protective effect by metformin. Some authors attributed the increase in lifespan thanks for its effects on cellular metabolism

In the case of resveratrol:

Nevertheless, there are some studies which suggest the lack of significant findings to use this agent to the beneficial metabolic effects in non-obese post-menopausal women and healthy obese individuals The protective effects against atherosclesoris only have action in individuals without high risk

In the case of rapamycin:

“Side effects have been described of chronic administration such as ulceration of mucosal tissues, haematological abnormalities, induction of insulin insensitivity, obesity, and diabetes, though these adverse effects may be largely dose-dependent.”

  1. In the abstract, the authors refer to HIV as a disease and refer to it as an event in the introduction. Neither is strictly correct with “HIV infection” contributing to physiological abnormalities that involve cellular senescence.

This has been corrected in the new version

  1. Much more could be made of the role of cytomegalovirus, contributing to immune senescence in its own right, and accelerating aging in HIV infection both through driving inflammation and driving immune senescence.

This has been considered in the new version.

...coinfection with other viruses (i.e., cytomegalovirus [CMV], Epstein-Barr virus, hepatitis B virus, and hepatitis C virus) is common in PLWH and coinfection with other viruses (i.e., cytomegalovirus) has been associated with inflammation and immune dysfunction [130]. One of the most important chronic co-infections is CMV, which has been associated with immune senescence, among others, which induced a chronic low-grade inflammatory state contributing to adverse health outcomes. Another reason could be the dysfunction of the intestinal barrier that allows bacterial translocation [131], promoting inflammation and immunoactivation (Figure 4). CMV can also aggravate intestinal epithelial damage, bacterial translocation and chronic intestinal inflammation.

  1. Line 45 appears grammatically incorrect (“inducers that mechanisms…”). Corrected
  2. Line 67 is grammatically incorrect. (“although” should be “through”?). Corrected
  3. On line 81, the abbreviation SAPS is used. I may have missed it, but couldn’t find the definition. Definition in line 64
  4. The meaning of lines 275-276 is unclear. (“in” should be “from”?)

Corrected

“Cellular stress and heat shock cause the activation of full-length HIV transcrip-tion-producing infectious virus from dormant HIV genomes”

Reviewer 3 Report

This review describes the current knowledge and challenges in applying senotherapeutics to treat cancer and HIV-1 disease. Although this is a timely and interesting topic, the part of the discussion related to HIV-1 is not complete. The manuscript lacks clarity and needs to be organized better; suggestion below.

Chapter 1. Biology and molecular mechanisms of senescence

Definition, functions, types, triggers, molecular pathways

There are three major types of senescence: replicative senescence (telomere dependent), programmed senescence (=developmental senescence), and stress-induced premature senescence (non-telomeric). Stress-induced premature senescence includes oncogene-induced (OIS), unresolved DNA damage-induced (genotoxin-induced), epigenetically induced, and mitochondrial dysfunction-induced. A common trigger for senescence induction is the activation of persistent DNA damage response (DDR).

Chapter 2. Role of senescence in cancer

Senescence is generally a tumor-suppressive process preventing cancer cell proliferation and suppressing malignant progression.

Deleterious effects of cancer therapy-induced senescence (in both tumor and healthy tissues).

Chapter 3. Senescence and HIV

HIV-associated chronic inflammation as a driver of immune senescence and acceleration of aging. T cell activation and exhaustion.

Chapter 4. Senotherapeutics.

Categories of senotherapeutics, Figure 3

The goals of senotherapy in cancer: 1) targeted removal of chemotherapy-induced senescent cells 2) blocking deleterious SASP. Table 1, Figure 2

The goals of senotherapy in HIV: 1) reducing immune cell activation and inflammation 2) clearance of latently infected cells. Table 2, Figure 4

Chapter 5. Conclusions and perspectives.

Expand Table 1:

Add in brackets navitoclax (ABT-263)

Add and describe:

Other pan-BCL2/XL inhibitors: obatoclax (GX15-070), A1331852, A115463,

Other mTOR inhibitors: temsirolimus, everolimus, AZD8055

Cardiac glycosides (senolytics): Digoxin, Ouabain,

Fenofibrate (senolytic)

HSP90 inhibitors (senolytics): Tanespimycin (17-AAG), Radicilol, Geldanamycin

Piperlongumine (senolytic)

UBX0101 (senolytic)

KU-60019 (senostatic)

Create Table 2 for senotherapeutics used for HIV-1, describing their effects on HIV-1 infected cells and HIV-1-related immune cell dysfunction.

Table 2. Senotherapeutics in HIV-1 disease.

Quercetin ® reducing ART-induced neuroinflammation, latency reversal

Venetoclax (ABT-199) ® block proliferation of latently infected cells, inducing apoptosis

Rapamycin ® regulating viral persistence

Ruxolitinib ® blocking HIV replication in macrophages

Dasatinib ® upregulation of SAMDH1 in infected macrophages, reduction activation and proliferation of neutrophils and T-cells

Other: Fenofibrate, Tanespimycin (17-AAG), Geldanamycin, Panobinostat

Clinical trials using senotherapeutics for HIV

Sirolimus (=Rapamycin) (NCT02440789)

Everolimus (NCT02429869)

Ruxolitinib (NCT02475655)

Other comments:

Line 37: senescence activation is not responsible for “HIV”, change to “HIV persistence” and/or “HIV-induced immune system exhaustion”, similarly replace “cancer development” with “cancer progression/relapse”.

Line 45: remove “mechanisms prematurely”

Line 67: remove “although” add a comma

Line 381: The statement: “To date, there are no ongoing clinical traials on the use of senotherapeutics in PLWH” is not valid; see above comments and check NCT02440789, NCT02429869, NCT02475655

Author Response

This review describes the current knowledge and challenges in applying senotherapeutics to treat cancer and HIV-1 disease. Although this is a timely and interesting topic, the part of the discussion related to HIV-1 is not complete. The manuscript lacks clarity and needs to be organized better; suggestion below.

Chapter 1. Biology and molecular mechanisms of senescence

Definition, functions, types, triggers, molecular pathways

There are three major types of senescence: replicative senescence (telomere dependent), programmed senescence (=developmental senescence), and stress-induced premature senescence (non-telomeric). Stress-induced premature senescence includes oncogene-induced (OIS), unresolved DNA damage-induced (genotoxin-induced), epigenetically induced, and mitochondrial dysfunction-induced. A common trigger for senescence induction is the activation of persistent DNA damage response (DDR).

“Cellular senescence activation can be triggered by a range of stress stimuli. The first senescence mechanism discovered was gradual telomere shortening in proliferating cells (telomere dependent), which impaired cell division. This type of senescence is known as replicative senescence. Telomere erosion serves as a mitotic clock, inducing cell senescence, whose activation prompts normal cells to enter a state of proliferation arrest. This replicative senescence does not occur in stem cells or tumor cells since they express elevated levels of telomerase, endowing them with unlimited replicative potential. Other types of cellular senescence include programmed senescence (developmental senescence), which is essential in embryo development, and stress-induced premature senescence (SIPS), which can be activated by a variety of non-telomeric stress signals. One well-established example of SIPS is oncogene-induced senescence (OIS), which was initially characterized when the Ras oncogene was overexpressed in primary mammalian cells and was later associated with other oncogenes. Different treatment regimens can also turn on SIPS, such as chemo-or radiotherapy, inducing a persistent DNA damage response (DDR) (genotoxic-induced) and a cancer cell proliferation blockade. In addition, other types of senescence have been described, such as epigenetically induced and mitochondrial dysfunction-induced senescence.”

Chapter 2. Role of senescence in cancer

Senescence is generally a tumor-suppressive process preventing cancer cell proliferation and suppressing malignant progression.

“Cancer is the result of uncontrolled proliferation of cells, which can sometimes invade into other tissues, causing metastasis. In cancer, senescent cells also have opposite effects. In the early stages, senescence acts as a tumor suppressor, decreasing cell proliferation in response to oncogene expression, preventing cancer cell proliferation and suppressing malignant progression. Additionally, senescent cells are able to attract immune cells to the tumor site, promoting the recognition and clearance of tumor cells by immune system.”

Deleterious effects of cancer therapy-induced senescence (in both tumor and healthy tissues).

“Indeed, in cancer patients treated with radiotherapies or chemotherapies, the induced senescence in tumor and healthy tissues has been related to stemness induction, relapses, metastasis and a worse outcome.”

Chapter 3. Senescence and HIV

HIV-associated chronic inflammation as a driver of immune senescence and acceleration of aging. T cell activation and exhaustion.

This has been considered in the new paragraph:

In the same way, coinfection with other viruses (i.e., cytomegalovirus [CMV], Epstein-Barr virus, hepatitis B virus, and hepatitis C virus) is common in PLWH and has been associated with inflammation and immune dysfunction [130]. One of the most important chronic co-infections is CMV, which has been associated with immune senescence, among others, which induced a chronic low-grade inflammatory state contributing to adverse health outcomes [33167724]. Another reason could be the dysfunction of the intestinal barrier that allows bacterial translocation [131], promoting inflammation and immunoactivation (Figure 4). CMV can also aggravate intestinal epithelial damage, bacterial translocation and chronic intestinal inflammation [28241080].

PLWH are exposed to multiple stressors that can induce a premature cellular senescence, a higher risk of comorbidities and HIV disease progression. In vivo and in vitro evidence has shown that HIV contributes to cellular senescence [132-135]. Therefore, in contrast to healthy controls, untreated PLWH have increased p16 INK4a levels [136]. Excessive accumulation of senescent cells in numerous tissues leads to multiple chronic diseases, tissue dysfunction, age-related diseases and organ aging [126]. Indeed, some studies have suggested that long-term ART might influence immune senescence in people living with HIV [137-140]. So, protease inhibitors, one of the main family of ART, inhibit the maturation of pre-lamina A into lamina A [14600514] which accumulation has been linked to age-related diseases (i.e., Hutchinson-Gilford progeria syndrome). Finally, Zhao et al. [26775705] also observed a strong association of cell senescence with HIV-1 infection and viral carcinogenesis.

Chapter 4. Senotherapeutics.

Categories of senotherapeutics, Figure 3

The goals of senotherapy in cancer: 1) targeted removal of chemotherapy-induced senescent cells 2) blocking deleterious SASP. Table 1, Figure 2

Corrected

The goals of senotherapy in HIV: 1) reducing immune cell activation and inflammation 2) clearance of latently infected cells. Table 2, Figure 4

 Corrected

Chapter 5. Conclusions and perspectives.

 The conclusions have been improved.

Expand Table 1:

Add in brackets navitoclax (ABT-263). Corrected

Add and describe:

Other pan-BCL2/XL inhibitors: obatoclax (GX15-070), A1331852, A115463,

“The increase of the selectivity could be improved the hemotological side effects caused by Navitoclax. For this reason, A1331852 and A115463 which are selective inhibitors of BCL-XL may be better candidates for translation into clinical applications. These two inhibitors act as in senescent HUVECs and IMR-90 cells but not primary human preadipocytes”

Other mTOR inhibitors: temsirolimus, everolimus, AZD8055

“Everolimus, an analogue of rapamycin, recruits the immunophilin/prolyly isomerase FKBP12 to mTORC1 and has better bioavailability and pharmacokinetics.”

“The second generation of mTOR inhibitors compete with ATP for the active site of Mtor kinase inhibiting both mTORC1 and mTORC2. AZD8055 presents high specifity and selectivity for mTOR kinase. The ATP-competitive inhibitors show more apoptotic effects in vitro than rapamycin or everolimus but in vivo have not yet demonstrated better efficacy than current treatment regimens

Cardiac glycosides (senolytics): Digoxin, Ouabain,

“Cardiac glycosides

Senescent cells present a little plasma membrane depolarization and higher level of H+. Through the Na+/K+ATPase pump, cardiac glycosides (Digoxin and Ouabain) induce a dysregulation in electrochemical gradient inside the cells causing depolarization and acidification. Furthermore, the treatment with ouabain or digoxin promoted apoptosis in senescent cells caused in part by the increase in the level of some proapoptotic BCL2 family proteins and NOXA. Hence, senescent cells could be more susceptible to the action of these agents. The activity as a senolytic was tested in vitro and in vivo, it is effective against tumor and primary cells. Despite, cardiac glycosides could be employed as anti-cancer drugs because of the immunogenic cell death and recent studies show the reduction of atherosclerosis and bleomycin induced pulmonary fibrosis.”

Fenofibrate (senolytic)

Described in subsection other senolytics

“Other senolytics

  • Fibrate: It was identified Fenofibrate (FN), a PPARα agonist used for dyslipidaemias in humans, as a senolytic. This agent induced apoptosis of senescent cells, increased autophagy and protected the cartilage degradation. Fibrate treatment improved osteoarthritis in patients from Osteoarthritis Initiative (OAI) Cohort in a retrospective study”

HSP90 inhibitors (senolytics): Tanespimycin (17-AAG), Radicilol, Geldanamycin

Already described

Piperlongumine (senolytic)

Already described

UBX0101 (senolytic)

Described in subsection other senolytics

“UBX0101 is a p53/MDM2 interaction inhibitor. However, the mechanism to induce senescent cell apoptosis have not been fully elucidated. Intra-articular injection of UBX0101 selectively caused the clearance of senescent cells that are accumulated in the articular cartilage. More beneficial effects were the reduction in the development of post-traumatic osteoarthritis and the increase in the chondrogenesis. There are some clinical trials to test in knee arthritis (NCT04129944, NCT04349956, NCT04229225, NCT03513016).”

KU-60019 (senostatic)

 Described in subsetion ATM inhibitor

“ATM inhibitor

ATM (Ataxia-telangiectasia mutated) kinase inhibitor, KU-60019, was described as a senostatics. The activity of this drug caused the functional recovery of the lysosome/autophagy system, mitochondrial function and metabolic reprogramming. So that, senescence could have other mechanisms regulators, such as the lysosomal-mithochondrial axis modulated by ATM activity”

Create Table 2 for senotherapeutics used for HIV-1, describing their effects on HIV-1 infected cells and HIV-1-related immune cell dysfunction.

Table 2. Senotherapeutics in HIV-1 disease.

Quercetin ® reducing ART-induced neuroinflammation, latency reversal

Venetoclax (ABT-199) ® block proliferation of latently infected cells, inducing apoptosis

Rapamycin ® regulating viral persistence

Ruxolitinib ® blocking HIV replication in macrophages

Dasatinib ® upregulation of SAMDH1 in infected macrophages, reduction activation and proliferation of neutrophils and T-cells

Other: Fenofibrate, Tanespimycin (17-AAG), Geldanamycin, Panobinostat

Clinical trials using senotherapeutics for HIV

Sirolimus (=Rapamycin) (NCT02440789)

Everolimus (NCT02429869)

Ruxolitinib (NCT02475655)

 Table 3 has been created and included.

Other comments:

Line 37: senescence activation is not responsible for “HIV”, change to “HIV persistence” and/or “HIV-induced immune system exhaustion”, similarly replace “cancer development” with “cancer progression/relapse”.

 Corrected

Line 45: remove “mechanisms prematurely”

 Corrected

Line 67: remove “although” add a comma

 Corrected

Line 381: The statement: “To date, there are no ongoing clinical traials on the use of senotherapeutics in PLWH” is not valid; see above comments and check NCT02440789, NCT02429869, NCT02475655

Thanks the reviewer for calling our attention on this. Text has been corrected.

Reviewer 4 Report

In the study, “Senotherapeutics in Cancer and HIV” by Sanchez-Diaz, et al, the authors describe senescence, it’s role in disease, and the possibility of treating senescence with senolytics or senostatic drugs.  While generally easy to read, the authors really need to emphasize the senescence role of these drugs, as opposed to their known and established roles in anti-tumor, anti-viral, and anti-inflammatory pathways.  It is unclear how these drugs specifically target senescence in the diseases described in this manuscript.  Furthermore, a table describing these senolytic/senostatic drugs (outside HIV) could be added to further shed light on their role in senescence.   

Major Specifics:

Line 62:  What makes SASP the most important attribute of senescent cells, over other morphological and functional changes?  I think more time needs to be spend clearly demonstrating what SASP and SIPS entails, especially since these terms are used repeatedly in figures and throughout the manuscript.  What are SASP factors (line 146)? Perhaps a chart or figure outlining SASP factors would be useful.

Line 98:  Expand upon “among other examples”

Line 125:  It is not clear how this relates back to Figure 1.

Line 204:  “The main characteristics of senescent cells are metabolic shifts, epigenetic changes, and resistance to apoptosis…”  it feels like the characteristics and markers for senescent cells are described at multiple places in the manuscript.  It would be better to have one section that solely describes senescent markers and how these can be distinguished from other pathways (such as tumorigenic signals like NF-kb, IL6, growth factors and metastasis signals). 

Can the authors comment on the anti-senescent properties of drugs vs their anti-tumorigenic or anti-inflammatory properties? And are they separate or complimentary functions?  For example, Dasatinib and navitoclax have been used mainly for their anti-tumorigenic properties in fighting cancer.  Most of those studies did not evaluate senescent cells, but instead looked at endpoints such as the clearance of the tumor, the anti-proliferative functions of the drugs, and disease remission. 

Line 259:  Has covid-19 been linked to senescence?  It is unclear in some of the descriptions of these drugs as to whether they are being used to specifically target senescent cells or are being used to decrease inflammation, inhibit tumor cell growth, etc. 

Line 273:  The mechanism of action geldanamycin and similar drugs is the inhibition of heat-shock activity which protects cellar proteins from degradation.

Is also not clear how Hsp90 inhibitors used to treat HIV patients is linked to senescence. 

Line 312:  Unless you go into more detail (and define progerin, HGPS, and MRC-5 cells) this is too specific for the review.  Treatment with ruxolitinib rescued truncated lamin A (progerin)-induced cellular senescence, SASP in cultured MRC-5 cells and HutchinsonGilford progeria syndrome (HGPS)-derived fibroblasts….

Section 6 on HIV is great in describing inflammation and other barriers surrounding chronic HIV patients. However, more needs to be written and speculated on the role of senescence in long-term care.  The drugs listed in the table are not described.  Why are these drugs considered senolytic and senostatic?  Where is the evidence?  Since this is a review on senesce and drug therapies -more time needs to be spent describing the actual senescence (as opposed to inflammation, etc) and how drugs can specifically target this.  Furthermore – the table needs to list in a column what evidence is available that they are senolytic or senostatic. 

Minor Changes:

Line 40:  The use of the term “replicative initiation signaling” is not clear as written.

Line 44:  In addition to this replicative initiation signaling, there are other currently known inducers that mechanisms prematurely activate cellular senescence. Sentence not clear.

Line 81: “and showing antifibrotic functions” is not clear as written.

Line 137:  This sentence (Despite the current difficulties in well-known studying cellular senescence…) is not clear as written.

Line 166:  “for instance, chemo- and radiotherapy side effects due to not being targeted therapies”…needs clarity.

Line 222:  What is the ATPLite assay?

Line 276:  These paragraphs do not flow.  You already introduced Hsp90 at the beginning of this paragraph and are now re-defining Hsp90 later… (the molecular chaperone heat shock protein 90 (Hsp90) – should be placed at beginning of paragraph, when you 1st introduce Hsp90). 

Author Response

In the study, “Senotherapeutics in Cancer and HIV” by Sanchez-Diaz, et al, the authors describe senescence, it’s role in disease, and the possibility of treating senescence with senolytics or senostatic drugs.  While generally easy to read, the authors really need to emphasize the senescence role of these drugs, as opposed to their known and established roles in anti-tumor, anti-viral, and anti-inflammatory pathways.  It is unclear how these drugs specifically target senescence in the diseases described in this manuscript.  Furthermore, a table describing these senolytic/senostatic drugs (outside HIV) could be added to further shed light on their role in senescence.   

Major Specifics:

Line 62:  What makes SASP the most important attribute of senescent cells, over other morphological and functional changes?  I think more time needs to be spend clearly demonstrating what SASP and SIPS entails, especially since these terms are used repeatedly in figures and throughout the manuscript.  What are SASP factors (line 146)? Perhaps a chart or figure outlining SASP factors would be useful.

Line 62: “Arguably, probably the most remarkable attribute of senescent cells is the senescence-associated secretory phenotype (SASP), due to its implication in the microenvironment and its important effects on a variety of diseases pathology. This characteristic involves the secretion of cytokines, chemokines, proteases, growth factors, extracellular media (ECM) elements and ECM-degrading enzymes. These molecules are described as influential in their microenvironment via autocrine and paracrine senescence stimulation….”

Line 161: “To unequivocally identify senescent cells, the most common markers used are a com-bination of SA-β-gal, lipofuscin, loss of nuclear high-mobility group box 1 (HMGB1) or lamin B1, increased levels of cell cycle inhibitors (i.e., p16INK4A) and p21 or SASP factors, such as cytokines, chemokines, proteases, growth factors, extracellular media (ECM) elements and ECM-degrading enzymes (Tabla 1).”

“Table 1. SASP factors.” has been created and included.

Line 98:  Expand upon “among other examples”

“On the other hand, there is evidence of persistent senescent cells in vivo. For instance, young adult women cured of breast cancer through treatment with cytotoxic chemotherapy exhibit increased expression of markers of cellular senescence for decades [38]. Other studies suggest that commonly used chemotherapies can induce persistent senescent cells in non-cancerous mice tissue. In addition, research on cancer survivors showed that one long-term effect of chemotherapy is the accelerated development of a variety of age-associated diseases.”

Line 125:  It is not clear how this relates back to Figure 1.

Corrected.

Line 204:  “The main characteristics of senescent cells are metabolic shifts, epigenetic changes, and resistance to apoptosis…”  it feels like the characteristics and markers for senescent cells are described at multiple places in the manuscript.  It would be better to have one section that solely describes senescent markers and how these can be distinguished from other pathways (such as tumorigenic signals like NF-kb, IL6, growth factors and metastasis signals). 

Corrected

The characteristics appeared only in the subsection correspondence (1.2). The senescent markers are described in therapy difficulties (section 4).

Can the authors comment on the anti-senescent properties of drugs vs their anti-tumorigenic or anti-inflammatory properties? And are they separate or complimentary functions?  For example, Dasatinib and navitoclax have been used mainly for their anti-tumorigenic properties in fighting cancer.  Most of those studies did not evaluate senescent cells, but instead looked at endpoints such as the clearance of the tumor, the anti-proliferative functions of the drugs, and disease remission. 

The drugs which appear in this review have anti-tumorigenic and/or anti-inflammatory effects. The clinical uses approved to FDA for these drugs are from solid tumors and hematological malignances to diabetes and inmunosupression. Moreover, some of them are being tested in experimental or differents phases of clinical trials to treat age-related diseases such as epigenetic aging, frailty and osteoarthritis, among others. All of these information could find in table 1 in this manuscript.

Line 259:  Has covid-19 been linked to senescence?  It is unclear in some of the descriptions of these drugs as to whether they are being used to specifically target senescent cells or are being used to decrease inflammation, inhibit tumor cell growth, etc. 

Senescent cells promotes inflammation, multile chronic disease and age-related diaseases. Recent studies tested that cellular senescence are hyper-inflammatory in response to pathogen-associated molecular patterns (PAMPs), such as SARS-CoV-2 spike protein-1. For this reason, the senotherapy could be helpful to treat this infection and there are clinical trial son going to use in this virus infection.

Reference: Camell CD, Yousefzadeh MJ, Zhu Y, Prata LGPL, Huggins MA, Pierson M, Zhang L, O'Kelly RD, Pirtskhalava T, Xun P, Ejima K, Xue A, Tripathi U, Espindola-Netto JM, Giorgadze N, Atkinson EJ, Inman CL, Johnson KO, Cholensky SH, Carlson TW, LeBrasseur NK, Khosla S, O'Sullivan MG, Allison DB, Jameson SC, Meves A, Li M, Prakash YS, Chiarella SE, Hamilton SE, Tchkonia T, Niedernhofer LJ, Kirkland JL, Robbins PD. Senolytics reduce coronavirus-related mortality in old mice. Science. 2021 Jul 16;373(6552):eabe4832. doi: 10.1126/science.abe4832. Epub 2021 Jun 8. PMID: 34103349; PMCID: PMC8607935.

Line 273:  The mechanism of action geldanamycin and similar drugs is the inhibition of heat-shock activity which protects cellar proteins from degradation.

Corrected

Is also not clear how Hsp90 inhibitors used to treat HIV patients is linked to senescence. 

Corrected

Line 312:  Unless you go into more detail (and define progerin, HGPS, and MRC-5 cells) this is too specific for the review.  Treatment with ruxolitinib rescued truncated lamin A (progerin)-induced cellular senescence, SASP in cultured MRC-5 cells and HutchinsonGilford progeria syndrome (HGPS)-derived fibroblasts….

Corrected

Section 6 on HIV is great in describing inflammation and other barriers surrounding chronic HIV patients. However, more needs to be written and speculated on the role of senescence in long-term care.  The drugs listed in the table are not described.  Why are these drugs considered senolytic and senostatic?  Where is the evidence?  Since this is a review on senesce and drug therapies -more time needs to be spent describing the actual senescence (as opposed to inflammation, etc) and how drugs can specifically target this.  Furthermore – the table needs to list in a column what evidence is available that they are senolytic or senostatic. 

 Drugs are described and new table 2 added  .

Table2 : Some of the main senolytic tested in HIV

Minor Changes:

Line 40:  The use of the term “replicative initiation signaling” is not clear as written.

Corrected

Line 44:  In addition to this replicative initiation signaling, there are other currently known inducers that mechanisms prematurely activate cellular senescence. Sentence not clear.

Corrected

Line 81: “and showing antifibrotic functions” is not clear as written.

“Additionally, its activation during specific periods of embryonic and adult life is crucial for development, ensures stressed and damaged cell clearance by immune cells and exerts an antifibrotic function.”

Line 137:  This sentence (Despite the current difficulties in well-known studying cellular senescence…) is not clear as written.

“Despite the current difficulties in the study of cellular senescence in vitro and in vivo, different approaches have been implemented in research on the molecular biology and behavior of senescent cells.”

Line 166:  “for instance, chemo- and radiotherapy side effects due to not being targeted therapies”…needs clarity.

“Furthermore, conventional therapy for the disease might show important disadvantages (for instance, chemo- and radiotherapy side effects in non-tumoral tissues) that could be eliminated with the addition of or replacement by senotherapy.”

Line 222:  What is the ATPLite assay?

ATP is a marker for cell viability because it is present in all metabolically active cells. The concentration of ATP decline fastly when cells undergo necrosis or apoptosis. ATPlite™ is an Adenosine TriPhosphate (ATP) monitoring system based on firefly (Photinus pyralis) luciferase. This luminescence assay does a quantitative evaluation of proliferation and cytotoxicity of cultured mammalian cells.

Line 276:  These paragraphs do not flow.  You already introduced Hsp90 at the beginning of this paragraph and are now re-defining Hsp90 later… (the molecular chaperone heat shock protein 90 (Hsp90) – should be placed at beginning of paragraph, when you 1st introduce Hsp90)

Corrected

Round 2

Reviewer 4 Report

The authors have improved the manuscript; and scientifically it reads better.  However, the English needs extensive improvements. This is especially true of the recently added portions.  For example:

Line 67: Arguably, probably the most important remarkable... Does not make sense

Line 379:    Rapamycin caused senescence through a nuclear factor E2-related factor 2 Nrf2-independent mechanism...  This is the opposite!  According to the rest of the paragraph, rapamycin decreased senescence.

Line 394:  All use of In vitro/in vivo should be italized

Line 411:  Two periods are used at the end of a sentence

Lines 416: The mainly doubt is if healthy subjects  - does not make sense

Lines 431, 437/438 - Unclear

Sometimes words are in bold for no reason.

There are too many sentences (especially the newly added "red" lines) that need extensive proofing.  Please have the manuscript professionally edited.

Author Response

Dear Sr,

English has been extensively edited by english native editors. corrections are highlighted in the "...tracked changes" PDF file. We hope the final version is much better  and suitable for publication. Also, correct text format has been implemented.

Best